# The Role of Urban Manufacturing for a Circular Economy in Cities

**Tanya Tsui** [1,2,*] , **David Peck** [1] , **Bob Geldermans** [1] and **Arjan van Timmeren** [2]

1   Architectural Engineering and Technology, Architecture and the Built Environment, Delft University of Technology, 2628 CE Delft, The Netherlands; D.P.Peck@tudelft.nl (D.P.); R.J.Geldermans@tudelft.nl (B.G.)
2   Urbanism, Architecture and the Built Environment, Delft University of Technology, 2628 CE Delft, The Netherlands; a.vantimmeren@tudelft.nl
*   Correspondence: t.p.y.tsui@tudelft.nl

**Abstract:** In recent years, implementing a circular economy in cities (or "circular cities") has been proposed by policy makers as a potential solution for achieving sustainability. One strategy for circular cities is to reintroduce manufacturing into urban areas (or "urban manufacturing"), allowing resource flows to be localized at the city scale. However, the extent to which urban manufacturing contributes to circular cities is unclear in existing literature. The purpose of this paper is therefore twofold: to understand whether urban manufacturing could contribute to the circular economy, and to understand the drivers and barriers to circular urban manufacturing. By reviewing existing literature and interviewing experts, we identified the caveats for the contribution of urban manufacturing to circular cities, as well as the spatial, social, and material-related drivers and barriers for circular urban manufacturing.

**Keywords:** circular economy; circular cities; urban manufacturing; drivers; barriers

## 1. Introduction

Cities have a large environmental impact—they consume 60–80% of natural resources globally, produce 50% of global waste, and 75% of greenhouse gas emissions [1]. Reducing emissions and waste will be a major challenge for cities, and in recent years, transitioning to a circular economy has been proposed by policy makers as a potential solution [2].

While there is no common definition for the circular economy, it is generally understood as a closed-loop system that employs circular processes such as reuse, refurbishing, remanufacturing, and recycling to convert waste into resources [3].

To implement a circular economy at a city level, one proposed approach is to encourage local manufacturing in cities, minimizing the importation of raw materials and reliance on global supply chains [4]. This strategy seems to be gaining interest among practitioners and researchers—EUROPAN, a well-known and large-scale annual competition open to architects, urbanists, and landscape architects, had the topic of 'productive cities' for 2 years in a row. Moreover, a variety of on-going research projects on this topic is being funded by the European Commission. This includes Pop-Machina (which is funding this research), Reflow, and Centrinno.

At the same time, the topic "urban manufacturing" is being studied outside the field of circular economy, by scholars from urban planning, local economic policy, and manufacturing studies. These researchers explore the potential of reintroducing manufacturing into urban areas, by leveraging the availability of affordable, digital, and distributed production technology [5–9].

Although research on circular cities and urban manufacturing both study the localization of manufacturing in cities, exchange between the two fields is limited. Circular cities literature focuses on localizing material flows, but neglects the drivers and barriers for implementing urban manufacturing in cities. Urban manufacturing literature identifies

drivers and barriers, but has a limited understanding on the environmental impact of urban manufacturing processes.

Moreover, there is limited evidence that local urban manufacturing could contribute to a city level circular economy. The growing interest on this topic seems to be based on the assumption that urban manufacturers utilize local supply chains, decrease transportation emissions, and improve the environmental impact of the production process.

This paper therefore aims to question these assumptions by exploring whether urban manufacturing is indeed a viable alternative to centralized manufacturing when it comes to implementing a circular economy in cities. This paper will answer the following two research questions:

**RQ1:** *Does urban manufacturing contribute to a circular economy in cities, and if so, how?*
**RQ2:** *What are the drivers and barriers to circular urban manufacturing?*

By reviewing existing literature and interviewing experts on urban manufacturing and circular economy, we found that, while urban manufacturing contributes to a circular economy in cities, these claims come with a number of caveats, including the lack of empirical evidence, the relative insignificance of transportation emissions in the production process, and the continued reliance on global supply chains. With these caveats in mind, this paper then gives a definition of "circular urban manufacturing", and summarizes and categorizes its common drivers and barriers.

## 2. Background

### 2.1. Circular Economy in Cities

2.1.1. Circular Economy and Sustainability

Before introducing the theoretical background on circular economy in cities, a clarification on the relationship between the concepts of "circular economy" and "sustainability" is needed. While there appears to be connections between the two concepts, the similarities, differences, and relationships between the two remain ambiguous.

The most commonly accepted definition of sustainability is provided by the Brundtland Commission, stated as "development that meets the needs of the present without compromising the ability of future generations to meet their own needs" [10]. Circular economy, on the other hand, can be defined as "a closed-loop system that employs circular processes such as reuse, refurbishing, remanufacturing, and recycling to convert waste into resources" [3]. Its most important theoretical influences include cradle-to-cradle [11], looped and performance economy [12], and industrial ecology [13].

The difference between the concepts of sustainability and circular economy are two-fold. Firstly, the two concepts differ in terms of scope. On one hand, sustainability is focused on the so-called "triple bottom line" [14], and the three pillars of sustainability: people, profit, and planet. Literature on sustainability tends to focus on the "planet" pillar, measuring the environmental impact of activities using tools such as Life Cycle Assessments (LCA) and Material Flow Analysis (MFA). On the other hand, circular economy seems to have a stronger focus on the "profit" pillar, with literature dominated by a business-focused narrative aiming at profit-generating solutions, often in the form of business models. As a result, some authors argue that other dimensions, especially the social one, are not well integrated into the circular literature [15,16].

Secondly, the two concepts differ in terms of aims: while the main aim of a circular economy is to minimize the use of primary raw materials and waste in a production system by extending, intensifying, and closing material loops; sustainability addresses a multitude of issues such as greenhouse gas (GHG) emissions, land use, biodiversity loss, or toxicity, which may be prioritized differently according to the interest of researchers.

In a systematic literature review, Geissdoefer et al. (2017) categorized the relationship between CE and sustainability into three main types: 'conditional', 'beneficial', and 'trade-off'. The first type, a 'conditional' relationship, states that a circular economy is a necessary condition for a sustainable system [17,18]. Within this category, some authors argue that circular economy is one of the necessary conditions for a sustainable system [19], while

others argue that circular economy is the main solution [20]. The second relationship type, a 'beneficial' relationship, states that circular economy is beneficial for fostering a sustainable system [21,22]. Within this category, most authors argue that circular economy is not a necessary condition for fostering a sustainable system, but one of several solutions to do so [21,22]. The third relationship type, a 'trade-off' relationship, presents a more critical view of circular economy strategies. Authors argue that circular economy can have both costs and benefits in regard to sustainability, and could potentially lead to negative outcomes [22,23]. For example, some authors warn about the potential for circular systems to worsen the emission of greenhouse gases and accelerate global warming [23].

This research takes the perspective of a circular economy having a 'beneficial' relationship with sustainability, meaning it is one of several potential solutions for fostering a sustainable system. For this research, a circular economy is therefore not a system that closes resource loops for its own sake. Instead, the goal of circular resource flows is to achieve sustainability.

Moreover, by finding the drivers and barriers for (re)introducing circular urban manufacturing, this paper is attempting to examine the interaction between the micro and macro scales of the circular economy [3]. In other words, we are trying to articulate the relationship between city-scale circular strategies and individual urban manufacturers. This paper will have a focus on both techno cycle strategies, such as reuse, refurbish, remanufacture, and recycle; and biological cycles such as food production.

### 2.1.2. Why Circular Cities?

There are compelling arguments in literature for the potential of cities to be major drivers of the circular economy. The density and diversity of stakeholders in cities aids collaborations in closing, connecting, and continuing resource loops, and allows for the creation of various agents, organizations, and networks, which is increasingly important in the transition to a circular society [24]. Waste collected at the city scale is at a large enough quantity to justify harnessing through urban mining [25]. The topic of "circular cities" has emerged recently, including research reports on circular cities published by municipalities [26–28], and academic papers [29–31].

### 2.1.3. Connecting Circular Cities with Urban Manufacturing

Literature on circular cities can be separated into three main perspectives: Space (urban planning), People (urban governance), and Flows (urban metabolism). The spatial (or urban planning) perspective investigates how urban planning and zoning strategies affect circular activity in cities [32,33]. The people (or urban governance) perspective investigates how municipalities and policy makers implement circular strategies at the city level [4,16,31]. The flows (or urban metabolism) perspective investigates the flows of materials and waste in a city, and how resource flows can be recirculated at the city level [34,35].

One strategy in circular cities literature is the localization of resource flows to minimize the importation of raw materials and production of waste [4,36–39] by employing various circular activities. Strategies for developing circular economy in cities divides into two broad categories:

- Increasing production of products using locally grown raw materials [40,41];
- Increasing production or use of products using local secondary raw materials, which involves circular processes such as, refurbishing, remanufacturing, and recycling [42,43].

The majority (and most cited) articles on the localization of resource flows at the city level propose the implementation of eco-industrial parks [44–47]. However, literature is recently beginning to explore how circular economy can be implemented on the city as a whole, looking beyond eco-industrial parks and integrating industrial activity into urban areas. Rosado and Kalmykova (2019) developed a method to facilitate industrial symbiosis in the food industry in the municipality of Gothenburg, Sweden [35]. Mulrow et al. (2017)

explores industrial symbiosis opportunities at the scale of a single facility housing multiple firms, as an alternative to existing strategies for industrial parks [36].

However, while there is interest in the introduction of circular industrial activity into urban areas within circular cities literature, there is limited investigation into the drivers and barriers of implementing this. On the other hand, literature on "urban manufacturing", which investigates the (re)introduction of industrial activity into urban areas, focuses on principles that could potentially fit into a circular economy at the city level.

*2.2. Urban Manufacturing*

Research on urban manufacturing is motivated by a renewed interest of relocating manufacturing to urban areas. Technological developments allow manufacturing processes to be smaller, quieter, less polluting, and distributed, making it easier for manufacturers to justify their presence in cities [48–50]. The increased availability of cheap digital fabrication tools such as Computerized Numerical Control (CNC) routers, laser cutters, and 3D printers (also known as 'additive manufacturing'), as well as the increased presence of open workshops (such as fab-labs and makerspaces), has given more individuals and small businesses the opportunity to engage in urban manufacturing activity [5,7].

Urban manufacturing creates opportunities in local economic development [9], and some manufacturing businesses are also motivated to move back to urban areas to be closer to customers, business partners, consultancy services, and suppliers [48,51].

Urban manufacturing is studied under diverse disciplines and perspectives, and thus falls under a variety of different names. Under the term "industrial urbanism", urban planners study how urban form affects the development of industry in cities, and how new technological developments create the potential for new forms of industry in urban areas [8,49]. Under the term "urban manufacturing", policy researchers examine the drivers and barriers for urban manufacturing companies, as well as their potential for local economic development [9,48,52]. Under the term "distributed manufacturing" and "re-distributed manufacturing", designers study the priorities and capabilities of maker communities, as well as their potential for contributing to sustainability [53–56].

Some researchers go one step further, exploring how makerspaces can proliferate throughout a city, allowing for more independence from global supply chains [7,57]. This research is connected to various initiatives, including: The Fab City initiative (https://fab.city/), which is a global initiative for locally productive cities that originated in Barcelona; and Maker City (https://makercity.com/), which is an initiative that originated from the US, in response to the increasing popularity of the maker movement.

The definition of urban manufacturing varies across literature, and there is limited consensus on which types of production can be categorized as urban manufacturing. The main discrepancies of the definitions are due to scale of production. While some articles take a broader view of urban manufacturing and include craftsmen engaged in batch production [9], other articles on the topic only focus on manufacturers that operate at an industrial scale [6].

Fab City (fab.city), a global initiative promoting locally productive cities, provides a clear framework for different types of urban manufacturing, roughly categorized according to their scale of production. The framework is summarized below, in Figure 1. The different types of manufacturing are:

- **Personal fabricators:** hobbyists (sometimes referred to as "makers") making products for personal use.
- **Maker spaces:** (examples of makerspaces/fab labs can be found on: http://fablab.org/, https://artdesignxchange.com/) workshops where makers share fabrication space, equipment, and ideas.
- **Mini-factories**: (examples of mini factories can be found on: https://make.works/, https://madeinnyc.org/, https://www.urbanmfg.org/) small- to medium-sized manufacturing companies that have less than ~20 employees.

- **Traditional urban industry:** large-scale manufacturers that have chosen to stay in the city instead of offshoring production.

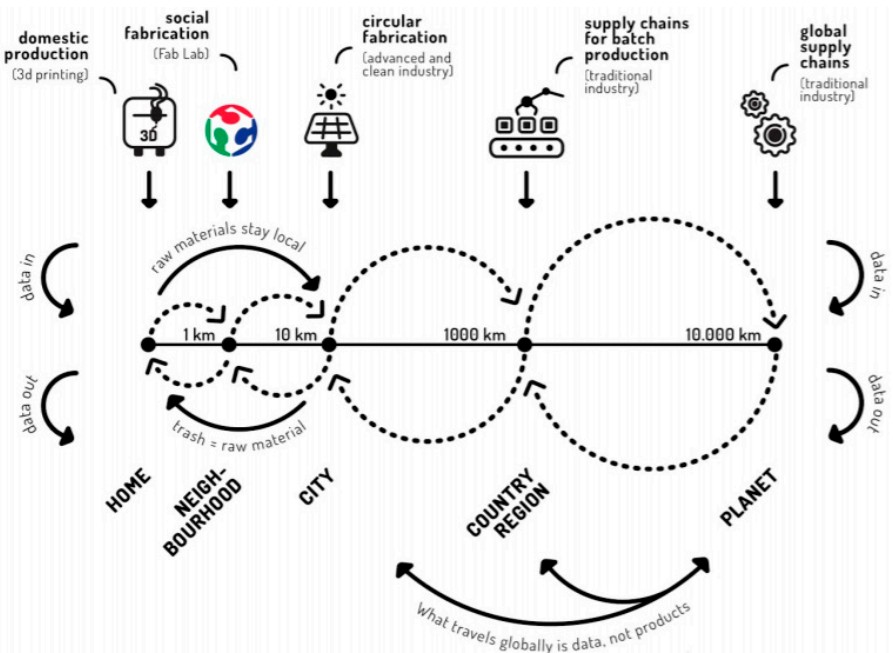

**Figure 1.** A multi-scalar and complimentary fabrication ecosystem [57].

This paper will focus on urban manufacturing activity at the scale of makerspaces and mini-factories, and for consistency, all manufacturing in cities will be referred to as "urban manufacturing".

Urban Manufacturing and Circular Economy in Cities

While there is existing research on the effects of urban manufacturing on local economic development, there has been limited exploration of how urban manufacturing could contribute to a circular economy in cities. In the existing literature, the potential environmental benefits of urban manufacturing are based on the following claims: that local supply chains can reduce transportation emissions [58–60], and that urban manufacturers can utilize local waste flows as a resource [16,29,33,61]. However, most of these claims are not supported by empirical evidence [54].

Given the lack of connection between urban manufacturing and circular economy, this paper aims to provide a deeper understanding of whether urban manufacturing contributes to a circular economy at the city level. In addition, drivers and barriers to urban manufacturing in a circular economy will be identified.

## 3. Materials and Methods

The aim of this research was to establish the extent to which urban manufacturing contributes to developing a circular economy in cities. This paper therefore attempted to answer two research questions: (1) Does urban manufacturing contribute to a circular economy in cities, and if so, how? and (2) What are the drivers and barriers to circular urban manufacturing?

The research questions were answered using three sources of information: a literature review of Life Cycle Assessments (LCAs) of urban manufacturing processes, semi-structured interviews of experts within the fields of circular economy and urban manufacturing, and a literature review of drivers and barriers in both circular cities and urban manufacturing.

### 3.1. Literature Review on Life Cycle Assessments (LCAs) of Urban Manufacturing

It was decided that a literature review of LCAs was appropriate for this research because LCA is a widely accepted and standardized methodology for assessing environmental impacts of products, processes, and services. The rigorous nature of LCAs allows for a relatively reliable comparison between different production scenarios, such as manufacturing using local versus global supply chains, or manufacturing for a local versus global consumer base.

For the literature review of life cycle assessments (LCAs) of urban manufacturing processes, the Scopus search engine was used to search for peer-reviewed articles that conducted a life cycle analysis on the environmental impacts of urban manufacturing. The search terms used were: TITLE-ABS-KEY ("makerspace" OR "urban industry" OR "local industry" OR "distributed industry" OR "urban manufacturing" OR "distributed manufacturing" OR "local manufacturing" OR "urban production" OR "local production" OR "distributed production" AND "LCA" OR "environmental impact" OR "carbon emissions").

The resulting 63 document results were further narrowed down to 9 articles, which examined the environmental performance of different types of food and consumer products (for a summary of the literature review, please refer to the Supplementary Materials).

The articles were chosen based on the following criteria. Each article:

- conducted a life-cycle assessment that provided empirical evidence for the total GHG emissions during the production and transportation process.
- made a comparison between the environmental impact of urban (or distributed) manufacturing and traditional centralized manufacturing.

### 3.2. Semi-Structured Interviews with Experts in Circular Economy and Urban Manufacturing

Semi-structured interviews were selected as the method for this research because the conversational nature of semi-structured interviews allows for a deeper understanding of the topics explored, giving interviewees an opportunity to elaborate on relevant case studies and speculative ideas that cannot be found in literature. The interviewees, who were either practitioners or researched closely with practitioners, provided additional insights from the perspective of urban manufacturers, which is essential to understanding drivers and barriers.

For the interviews of experts within the fields of circular economy and urban manufacturing, eight interviewees were chosen based on their expertise in urban manufacturing and circular cities, as well as recommendations from previous interviewees. Interviewees included both academics and practitioners. Academics were chosen based on their authorship of relevant papers, and practitioners were chosen based on their involvement in major urban manufacturing initiative, such as Fab City and the Urban Manufacturing Alliance. The interviews were semi-structured, and conducted online. The list of interviewees is found in Table 1.

The interview questions were as follows:

1. What is your definition of circular cities?
2. What is your definition of urban manufacturing?
3. Could urban manufacturing be a driver for the circular economy?
4. Could circular economy be a driver to urban manufacturing?
5. Could urban manufacturing and circular economy be a barrier for each other?
6. Should manufacturing be situated in cities?
7. Where in cities should they be situated, and why?
8. What kinds of manufacturing is / is not suitable for cities?
9. In what condition should products be produced locally?
10. How do you see the future of manufacturing in cities?

**Table 1.** List of interviewees.

| Interviewee | Country | Affiliation | Role | Date |
|---|---|---|---|---|
| A | United States of America | Yale University | Academic on urban planning, urban planning history, urban manufacturing | 7 February 2020 |
| B | United States of America | City University of New York | Academic on urban economic planning, urban manufacturing, the maker movement | 12 February 2020 |
| C | United Kingdom | University College London | Academic on circular economy, energy | 3 April 2020 |
| D | Spain | Institute for Advanced Architecture of Catalonia | Academic and practitioner on urban manufacturing, urbanism, the maker movement, fab.city co-founder | 9 April 2020 |
| E | United States of America | Urban Manufacturing Alliance | Practitioner on urban manufacturing network building | 10 April 2020 |
| F | Belgium | Université libre de Bruxelles | Academic on urban manufacturing, urban planning | 21 April 2020 |
| G | United States of America | UPcyclers Network | Practitioner on circular economy, policy and business models for recycling and reuse | 26 May 2020 |
| H | United States of America | Square Nail Consulting | Practitioner on circular economy, building product reuse | 3 June 2020 |

Each interview, which lasted from 30 min to 1 h, was recorded, transcribed, and sent to interviewees for verification. The transcriptions were then coded using the Atlas.ti program, where key sentences describing drivers and barriers were highlighted and categorized as a series of ideas. This resulted in more than 200 ideas. These ideas were then categorized as a driver or barrier, and further sub-categorized as space, people, or flow-related. The resulting list was summarized in an Excel file, which can be found in the Supplementary Materials. A similar process was followed when coding the literature on drivers and barriers for circular urban manufacturing.

With insights from the literature review and expert interviews, it was then possible to establish whether and how urban manufacturing contributes to a circular economy at the city level. Urban manufacturing that contributes to a circular economy was defined within this paper, as "circular urban manufacturing". This definition allowed us to answer the second research question, defining the drivers and barriers for circular urban manufacturing.

*3.3. Literature Review of Drivers and Barriers to Circular Urban Manufacturing*

The second research question was answered with a literature review of drivers and barriers identified in both circular city and urban manufacturing literature. Drivers and barriers that apply to circular urban manufacturing were identified and categorized into three perspectives: space, people, and flows. These three perspectives of space, people, and flows were chosen because the factors that affect the presence of manufacturing in urban areas are multi-faceted. Not only is the presence of urban manufacturing dependent on the availability of materials and technology (flows), it is also dependent on spatial issues (such as the availability of industrial land), and people-related issues (such as the presence of a support ecosystem). Instead of conducting an in-depth exploration onto one perspective, this paper aimed to give readers a broad overview of issues connected to circular urban manufacturing.

The three perspectives took reference from the Ecopolis framework of Urbanist Sybrand Tjallingii, where he highlights a threefold strategy for ecologically sound development, focusing on 'sites', 'participants', and 'flows' [62].

This paper changed the wording of the framework, to 'space', 'people', and 'flows'. 'Space' refers to issues related to land-use, land prices, and proximity of stakeholders.

'People' refers to issues related to management, money, networking, and education. 'Flows' refers to issues related to material flows, supply chains, and logistics.

Literature Search, Selection, and Coding Process

For this literature review on drivers and barriers for circular urban manufacturing, the Scopus search engine was used to search for peer-reviewed articles on the drivers and barriers for circular cities and urban manufacturing.

The search terms are summarized in the Table 2.

**Table 2.** Search terms for literature review of drivers and barriers to circular urban manufacturing.

| Topic | Search Terms | # Results |
|---|---|---|
| Drivers and barriers for circular cities and urban manufacturing | TITLE-ABS-KEY ("circular economy" OR "industrial symbiosis" AND "makerspace" OR "urban industry" OR "local industry" OR "distributed industry" OR "urban manufacturing" OR "distributed manufacturing" OR "local manufacturing" OR "urban production" OR "local production" OR "distributed production" AND "urban" OR "city" AND "driver" OR "opportunity" OR "barrier" OR "challenge") | 4 |
| Drivers and barriers for circular cities | TITLE-ABS-KEY ("circular economy" OR "industrial symbiosis" AND "urban" OR "city" AND "driver" OR "opportunity" OR "barrier" OR "challenge") | 270 |
| Drivers and barriers for urban manufacturing | TITLE-ABS-KEY ("makerspace" OR "urban industry" OR "local industry" OR "distributed industry" OR "urban manufacturing" OR "distributed manufacturing" OR "local manufacturing" OR "urban production" OR "local production" OR "distributed production" AND "urban" OR "city" AND "driver" OR "opportunity" OR "barrier" OR "challenge") | 184 |

As seen in the table, searching for papers that covered both circular cities and urban manufacturing resulted in only 4 papers. Due to the limited results, two separate searches were conducted for circular cities and urban manufacturing. The initial 458 papers were further narrowed down to 31 papers. The papers were chosen with the following criteria. Each paper:

- Focused on listing out drivers and barriers for circular cities or urban manufacturing.
- Provided a general overview of drivers and barriers, rather than focuses on specific materials and products.
- Contained drivers and barriers that can be categorized as space, people, or flow related.

The selected 31 papers were then coded using the Atlas.ti program, using a similar process to coding the interviews. Key sentences describing drivers and barriers were highlighted and categorized as a space, people, or flow-related driver or barrier. The resulting list was summarized in an Excel file, which can be found in the Supplementary Materials.

## 4. Results

### 4.1. Does Urban Manufacturing Contribute to a Circular Economy at the City Level?

In order to understand whether urban manufacturing contributes to a circular economy at the city level, a literature review of LCAs of urban manufacturing was conducted. Typically, the LCA method takes into account a variety of different environmental impact categories, such as greenhouse gas (GHG) emissions, land use, toxicity, acidification, eutrophication, ozone depletion, and various other indicators. This research, however, focuses on GHG emissions as an indicator of environmental impact, because reducing GHG emissions is a major strategy for mitigating the environmental damage of global warming and climate change [63]. Moreover, GHG emissions is an indicator that is commonly used across most LCA studies, allowing for a more reliable comparison between different findings.

### 4.1.1. Empirical Evidence for Environmental Impact of Urban Manufacturing

In articles that conducted an LCA on urban manufacturing processes, authors found that, while urban manufacturing contributes to reducing GHG emissions, this claim comes with a number of caveats and conditions.

Using LCAs, a number of authors provided empirical evidence for the positive environmental impact of urban manufacturing. Authors found that shortening transportation distances reduces GHG emissions by 0.8–2.6%, although improving other parts of the production process had a more significant impact. Benis and Ferrão (2017) found that eliminating losses and wastage during the production processes reduced emissions by 8%, while Russell and Allwood (2008) found that urban manufacturing with recycled materials reduces emissions by 9.5% [64,65]. This is because, for most consumer products, the largest source of emissions comes from production of raw materials, not transportation.

Other authors also found that urban manufacturing reduced GHG emissions, not due to shorter transportation distances, but due to other aspects of the urban manufacturing process. Hall et al. (2014) found that localizing the production of food had a positive environmental impact, because local farmers were more environmentally conscious and used better fertilizers. M. Kreiger and Pearce (2013) found that distributed manufacturing of consumer products with 3D printers can reduce emissions, because they are more energy efficient, and 3D printed products use less materials [66,67].

However, authors found that urban manufacturing also creates changes in the production process that lead to a negative environmental impact. A number of authors found that, for some types of food, decentralized manufacturing can be less environmentally friendly because smaller manufacturers cannot take advantage the efficiencies of economies of scale [66,68].

Localizing manufacturing may also lead to a negative environmental impact because of the local context. For example, the local electricity grid may use fewer renewable sources [65], or local climate conditions lead to less efficient production of crops [69].

### 4.1.2. Caveats to the Circularity of Urban Manufacturing Found in Literature and Expert Interviews

From literature and expert interviews, it was found that there are a number of caveats to the claims that urban manufacturing contributes to a circular economy. While many articles have claimed that urban manufacturing contributes to a circular economy, most of these claims are based merely on potential benefits, and are not backed-up by empirical evidence [54]. Moreover, not all types of urban manufacturing contribute to a circular economy in cities.

Not all urban manufacturers source from local supply chains. While some urban manufacturers may start off their business by sourcing local materials from nearby suppliers, it is difficult to stay local, especially when production starts scaling up. For many urban manufacturers, relying on offshore supply chains or moving manufacturing completely offshore is the only way to scale up production. In many cases, local manufacturing networks simply cannot compete with global offshore networks when it comes to price, efficiency, and knowledge [48,52,70].

Not all urban manufacturers aim to serve a local consumer base. These manufacturing businesses (referred to as "global innovators" [9]) often manufacture high-tech products, and are located in cities in order to access highly skilled professionals, such as designers, engineers, academics, or consultants. The products produced by these manufacturers, such as specialized medical or aerospace equipment, have a consumer base that far exceeds the boundaries of the city [9]. An urban manufacturing expert states, "*of course there are manufacturers that just produce for the local population, but for most companies to compete in the marketplace, they can't limit where they sell. And with global commerce and free trade, you can sell anywhere (in the world)*" (Interviewee A).

The issue of limiting supply to a single city also applies to circular product life extension processes such as reuse and recycling. Expert interviewees stated the impracticality of recycling certain types of materials at the city-scale.

- *"In Belgium, there's just a couple of metal treatment plants. So anytime a building is being totally renovated or rebuilt, all that steel that comes out gets taken to Antwerp or Ghent to be recycled"* (Interviewee F).
- *"It's an issue of the product and the scale. If you have, let's say, small lithium batteries, you need quite a lot of them to make it worth setting up a recycling plant. You could imagine that you only need one plant for the whole of the UK"* (Interviewee C).
- *"Large recycling processors want to have tons (of waste) coming in monthly, because they're looking for a certain percentage of returns for their investors. I don't think it's economically viable for them to operate within a city"* (Interviewee G).

Even if urban manufacturers had a positive environmental impact, their contribution to the overall environmental impact of a city is insignificant, as cities are still reliant on global centralized manufacturers. Urban manufacturers often operate at a smaller scale compared to centralized global manufacturers due to spatial and financial constraints, or simply because they do not desire to scale up [9].

From the expert interviews and the literature, one of the main paradoxes in urban manufacturing literature is this: if urban manufacturers want to stay local, they must stay small, reducing their potential impact on the city. If they scale up and try to grow their business, their positive impact may increase, but they often leave the city completely.

After examining the caveats on the sustainability of urban manufacturing, it can be concluded that urban manufacturing would only contribute to the circular economy under a number of conditions. Therefore, for this paper, "circular urban manufacturing" can be defined as urban manufacturing processes where:

- The business sources from local supply chains, and produces for a local consumer base.
- Transportation emissions of the product being manufactured contributes to a significant percentage of the total environmental impact of the product (for example, products produced from secondary raw materials will have much lower emissions associated with material extraction and processing).
- Local waste or secondary raw materials is used as a resource (this includes materials from both technical and biological cycles).
- There is a possibility of scaling up without moving out of the city.

### 4.2. Drivers and Barriers to Circular Urban Manufacturing

Through a literature review and interviews with experts, the drivers and barriers to circular urban manufacturing can be derived. Since there is limited literature and experts that examine the overlap between the two topics, the drivers and barriers for circular cities and urban manufacturing were extracted separately. Then, drivers and barriers that were relevant to "circular urban manufacturing" (as defined in the previous section) were selected and summarized in the following section.

#### 4.2.1. Drivers

The drivers for circular urban manufacturing have been separated into two categories—"push" and "pull" factors. "Push" factors refer to the internal motivations of circular urban manufacturers, such as the potential benefits that could occur if circular urban manufacturing happens at a larger scale. "Pull" factors refer to external conditions from the surrounding context that create a fertile environment for circular urban manufacturing. In other words, the presence of pull factors in a city can give more opportunities for circular urban manufacturers to survive and thrive.

The push factors of circular urban manufacturing are explained in the paragraphs below, as space, people, and flow-related push factors.

In terms of push factors related to urban space, digitization of manufacturing has allowed production processes to operate at a smaller scale, justifying the presence manufacturing in urban areas, despite higher rental costs [70,71]. For urban planners, reintroducing urban manufacturing has the added benefit of place-making, by "connecting the means of production and tapping into the city's creative and constructive spirit" [49].

In terms of people-related push factors, advocates for urban manufacturing are motivated by the potential of reshoring manufacturing [70], which could promote local economic development and create local working-class jobs [9,49].

Increasing urban manufacturing can also lead to more independence from global supply chains. Cities with less urban manufacturers are arguably less resilient to disruptions in global supply chains [72]. Increasing local production could also allow producers to avoid negative externalities, which are often hidden in the complexity of global supply chains (Interviewee D).

Designers and manufacturers, on the other hand, are motivated by the fact that distributed manufacturing technologies create the opportunity for open and accessible manufacturing, as well as fast prototyping. Smaller and cheaper digital fabrication technologies make them more accessible, lowering the threshold of capital required to start a manufacturing business (Interviewee B). Organizational nimbleness and reduced prototyping costs allow designers and manufacturers to get their work into the public domain without too much upfront investment, allowing for a shorter and faster product development cycle [48,50,60,70].

In terms of flow-related push factors, urban manufacturing has the potential to contribute to a circular economy-shorter supply chains mean lower transportation emissions [54,55,60,70], and increased local manufacturing capacity gives a greater potential for turning local secondary or residual materials into local resources [29,31,58,70]. Moreover, the maker movement has a thriving repair, recycle, and upcycle culture, where, for example, additive technologies can facilitate the reparability of products [50,60].

Pull factors, which refer to external conditions from the surrounding context that create a fertile environment for circular urban manufacturing, are identified in the paragraphs below, and categorized into space, people, and flow-related pull factors.

In terms of space-related pull factors, authors found that the presence of urban manufacturing depends on the availability of affordable industrial land and manufacturing spaces. Municipalities' protective industrial zoning strategies have a positive effect on the presence of both circular and urban manufacturing activity. This is illustrated in the case study of urban manufacturing activity in Portland, USA [52], as well as in circular cities literature [16,43]. Protective industrial zoning was mentioned during interviews as well, "*if cities actually got serious about enforcing industrial zoning, and making sure that there were affordable production spaces, then I think you'd see a lot more small makers able to expand*" (Interviewee B).

Protective industrial zoning policies depend on the local government's ownership and control of land, which prevents private developers from converting industrial land into more profitable residential or commercial land [43,73]. This was also pointed out by an urban manufacturing expert, "*in hot market cities, cities are feeling tons of pressure to convert industrial land to housing or to other commercial uses like hospitality. There are advocates in those cities fighting to retain that industrial land so that those (manufacturing) jobs can stay there*" (Interviewee E). Additionally, interviewees point out that smaller declining towns with cheap real estate could have an advantage in revitalizing urban manufacturing (Interviewees B, G, H).

Space providers for makers, such as makerspaces and mission driven real estate developers, also increase urban manufacturing activity [58]. For example, New York's Greenpoint Manufacturing and Design Center and Brooklyn Navy Yard, Tillamook Station in Portland, and the Industrial Council of Near West Chicago are operated by mission-driven industrial landlords that take a double bottom-line approach to their rental properties [9].

In terms of people-related pull factors, disturbances in global supply chains have the potential to increase urban manufacturing activity. An interviewee uses the example of the COVID-19 global pandemic, *"A lot of cities are missing capacities to deal with (the pandemic), and to find materials for personal protective equipment. Cities like London and New York, that have kicked out their manufacturers, are now really depending on Fab Labs to produce these materials"* (Interviewee F).

Individual urban manufacturers locate in urban areas in order to be closer to existing customers and support networks. In order to compensate for higher rental costs, urban manufacturers target their products towards specific consumer markets that are willing to pay a higher price of urban manufactured goods. Customers of urban manufacturers include:

- Wealthy, environmentally conscious customers who are interested in locally-produced, design-driven, or customized products [49,51,70]; (Interviewees B, C)
- Design or technology driven companies, in sectors such as architecture, theatre, aerospace, that require customized manufacturing services [51,52] (Interview F, A, E)
- Niche markets, such as custom-made shoes, high-end bicycle messenger bags, or custom-made fire-fighter jackets [48,50]; (Interviewees A, B)

Experts have found that the existence of support networks aimed towards urban manufacturers contributes significantly to a thriving urban manufacturing sector in a city. Support networks are important to urban manufacturers because these businesses often operate at a smaller scale and at a higher risk. Stakeholders in support networks include:

- Large-scale traditional manufacturers that collaborate with makers in prototyping products or integrate makers into their production chain as sub-contractors [49,52,55,58]; (Interviewees C, E)
- Local production networks, which include local supply-chains of small-scale manufacturers, makerspaces which provide access to space and fabrication technology, as well as potential business partners and contractors [51,52]; (Interviewee F)
- Skilled workers and professionals [9,52,74]; (Interviewee A, C)
- Experts, consultants, and universities [49]; (Interviewee A)
- Marketing or business support, such as branding organizations [9]

In terms of flow-related pull factors, circular urban manufacturing is driven by the existing availability of municipal and industrial waste and secondary materials. There is a substantial accumulation of municipal waste in cities, as well as construction and demolition waste from buildings and infrastructure that have been either demolished or undergoing refurbishment. Moreover, new regulations such as China's Green Fence Operations in 2013 prevents large quantities of waste from being exported to developing countries. In the long term, this gives an opportunity for cities to recycle waste locally [31]. An expert interviewee referenced municipality-led efforts to *"encourage manufacturing companies to locate in the city to focus on municipal trash"* (Interviewee G). Proximity to the end user also provides opportunities to recapture valuable materials from products at their end of life [70].

Industrial waste, on the other hand, is usually higher in quality and quantity, which gives more opportunity for circular processes to happen at an industrial scale. An expert interviewee states that, *"from our research in London, we found that there is a huge amount of industrial waste, and it's relatively pure in the sense that it can be sorted relatively easily . . . there's a lot of capacity for the industrial sector manufacturers to be a lot more effective with their waste streams, so that's a real opportunity"* (Interviewee F).

Table 3 summarizes Section 4.2.1. by listing out the drivers for circular urban manufacturing, categorized into issues related to 'space', 'people', and 'flows'.

**Table 3.** Summary of drivers for circular urban manufacturing.

| Drivers for Circular Urban Manufacturing | | |
|---|---|---|
| **Space** | **People** | **Flows** |
| Push factors | | |
| Manufacturing is cleaner, quieter, and smaller, allowing manufacturing to move back into the city<br>Potential for place-making | Potential for reshoring manufacturing<br>Independence from global supply chain<br>Democratized manufacturing<br>Faster prototyping and product development | Potential for turning local waste to a local resource<br>Potential lower transportation emissions<br>Repair, recycle, upcycle culture in the maker movement |
| Pull factors | | |
| Cheap real estate in smaller declining towns<br>Availability of industrial land<br>Space providing stakeholders for makers | Disturbances to the linear global supply chains<br>Access to support networks<br>Existing consumer market for circular urban manufacturing | Availability of waste (municipal waste, industrial waste, waste not worth shipping to other countries) |

### 4.2.2. Barriers

The barriers for circular urban manufacturing are identified in the paragraphs below, and categorized as space, people, and flow-related barriers.

A major spatial barrier for both circular and urban manufacturing activity is the lack of industrial land in cities, which limits the availability of affordable spaces for both circular infrastructure (such as spaces for storage, collection, and recycling of materials) [31,33] and manufacturing spaces [9,48,52,58].

Researchers have observed that municipalities are allowing the conversion of industrial land into commercial and residential land to take advantage of higher property tax revenues. A global political shift towards neoliberalism has also led to the privatization of government-owned land, reducing municipalities' abilities to protect industrial land [31].

Even if urban manufacturers have non-polluting and quiet production processes, outdated land-use and zoning regulations prevent them from using non-industrial spaces [49]. When discussing zoning regulations, an expert on urban manufacturing stated, "*many of these zoning regulations are outdated. The big question now is how to create what's called 'performance zoning', whereby we can judge whether the factory is suitable for its urban location on a case by case basis, rather than having a blanket regulation*" (Interviewee A).

This raises the connection between urban planning and the development of urban manufacturing. Many European cities are converting their existing industrial areas into mixed living and working environments, with the hopes that some specific industries could continue to thrive. These efforts are not always successful—increased land values, nuisance complaints, and negative perceptions can drive manufacturers away from regenerated industrial districts. Thus, the scaling up of urban manufacturing depends heavily on the city development context.

Moreover, while cities may have land suitable for manufacturers, these areas can remain abandoned and under-used. Authors have studied this phenomenon under the term 'wastescapes', which includes areas in cities such as abandoned territories, underused areas, former industrial areas, and operational landscape and infrastructure for waste management. While wastescapes undoubtedly create negative impacts on surrounding areas, they also provide the possibility of creating a positive impact through regeneration. For example, regenerating these 'wastescapes' can help support circular concepts by incorporating land-use functions and facilities that help to close resource loops [75]. Allowing manufacturers to locate in wastescapes could partially increase the availability of affordable industrial land for urban manufacturers, as well as provide an opportunity to turn wastescapes into more productive and circular areas.

*4.3. People/Flows*

Although urban manufacturing may have social, economic, and environmental benefits, circular urban manufacturers often operate at a small scale, and have a limited impact. Circular urban manufacturers are often limited to producing products in luxury or niche markets, as they cannot compete in terms of price, volumes, and delivery schedules with globally produced products [9,70].

For urban manufacturers, higher sales prices of niche or luxury products compensates for higher rental costs [48]. Similarly, circular manufacturers in the repair or reuse industry are limited to collecting and reusing high-value waste, to take advantage of higher resale values. Expert interviewees cited examples for the reuse industry in construction materials: "*The only part of the industry that survived initially, were the people going for the high-end material-the Tiffany chandeliers, the doorknobs, the architectural millwork*" (Interviewee H).

Although urban manufacturers could theoretically have a greater positive impact by scaling up, there are many significant barriers which prevent them from doing so. Urban manufacturers often lack the resources and knowledge required to scale up their business.

In terms of access to resources, urban manufacturing firms lack access to capital and are not prioritized by investors. Without extra funding, firms find it difficult to invest in the technology or space required to scale up production. An expert on urban manufacturing states, "*There's a lot of venture capital running around looking for investment, but it has a bias towards immaterial things like software. Software or anything that has to do with tech is a magnet for angel investors and venture capital investors; whereas it's considered much more risky and kind of less 'sexy' to invest in a company that's making (physical) things*" (Interviewee B).

This lack of access to capital links to the need for powerful 'launching customers', such as governments or traditional industrial companies that could integrate mechanisms into their purchasing guidelines and place more emphasis on the value of urban manufacturers nearby.

Due to their small scale, urban manufacturers have limited access to both local and global production networks. Unlike traditional manufacturing companies, urban manufacturers have limited connections, making opaque supply chains difficult to navigate. Sub-contractors may also require a 'minimum order' that exceeds the production capacity of smaller firms [48].

With limited capacity and personnel, urban manufacturing firms lack the knowledge required to scale up their business, including knowledge in production management, high quality manufacturing, as well as business skills such as marketing and accounting [48]. An expert interviewee also noted that "*there's not as much technical assistance available, or orientation to business available to people who are manufacturing entrepreneurs*" (Interviewee B).

When urban manufacturers do manage to scale up their business, there is no guarantee that their production will stay in the city. When the scale of production increases, more production space is needed, making an urban location even more expensive. Sourcing from offshore networks becomes a better option, because production capacity, knowledge, and cheap services are more available in other countries [9,48,52].

Even if they stay in the city and source locally, successful urban manufacturers often get acquired by multinational firms that swiftly decide to move production offshore to countries with a lower labor cost. An urban manufacturing expert recalls a particularly successful manufacturer in Portland, "*they were really scaling up and moved to a bigger space, and then all of a sudden we read that they had been acquired by a multinational and they were leaving the area*" (Interviewee B).

Table 4 summarizes Section 4.2.2 by listing out the barriers for circular urban manufacturing, categorized into issues related to 'space', 'people', and 'flows'.

**Table 4.** Summary of barriers for circular urban manufacturing.

| Barriers for Circular Urban Manufacturing | | |
|---|---|---|
| **Space** | **People** | **Flows** |
| Outdated land-use and zoning regulations<br>Lack of space for circular infrastructure (such as storage, reuse, recycling)<br>Lack of affordable space for urban manufacturers | Lack of knowledge required to scale up business<br>Limited access to larger production networks<br>Limited access to venture capital -circular and manufacturing businesses are seen as high-risk investments | Scaling up production leads to sourcing from offshore networks<br>Circular urban manufacturers limited to working with high-value or niche products, which reduces total volume of waste converted into resources<br>Limited availability of quality recycled material from local sources |

## 5. Discussion

### 5.1. Significance of Results, Further Research Directions

The purpose of this paper was to identify the extent to which urban manufacturing could contribute to a circular economy in cities. Through reviewing existing literature and expert interviews, a number of caveats were revealed on the circularity of urban manufacturing. This section will identify and explain the significance of these caveats, as well as propose possible directions for further research.

#### 5.1.1. Geographical Scales of the Circular Economy

Our review of urban manufacturing LCAs has found that shortening transportation distances reduces GHG emissions by 0.8–2.6%, but improving other parts of the production process has a more significant impact. The relative insignificance of transportation emissions implies that reducing the geographical distance between suppliers, producers, and consumers may not significantly reduce the negative environmental impact of production processes.

This echoes the existing literature in the field of industrial ecology investigating the geographical scales of resource flows: here, researchers have found that localizing resource flows is often not the most effective strategy for sustainability, and that there is no 'ideal' scale for resources to be (re)circulated [76–79].

The implicit bias of circular cities literature towards local (as opposed to global) supply chains could be better understood [76]. Further research can be conducted on the conditions under which (re)circulating resources at a local scale is more preferable to a national or global scale. By understanding the conditions affecting the geographical scales of resource flows, we can identify what types of urban manufacturing could contribute to the circular economy, and under which conditions.

What was not discussed in this paper is the environmental impact of urban manufacturing on its immediate surroundings, or 'micro-impacts'. Although authors have made general statements on how new technology allows manufacturers to limit its nuisance on its neighbors nearby [8,49], there seems to be limited literature on the micro impacts of distributed manufacturing technology, such as the toxicity of 3D printer fumes [54], or health and safety impacts of having more delivery trucks in a neighborhood due to the presence of an urban manufacturer. If urban manufacturing will become more commonplace in the future, micro-impacts could potentially become a significant concern.

#### 5.1.2. Locally Embedded Urban Manufacturers

Literature on urban manufacturing seems to be partially motivated by the assumption that urban manufacturers are embedded 'locally' in the cities they are located in—that they make use of local suppliers, produce for a local consumer base, and contribute to local economic development—but we have found that this is not always the case.

There are two potential further research directions in response to this finding. Firstly, it would be beneficial to categorize urban manufacturers according to how 'locally embedded'

they are in terms of material use-whether they utilize local, national, or global supply chains; and whether they sell to a local, national, or global consumer base. Wolf-Powers (2017) has made an excellent categorization of urban manufacturers in the US according to their potential for local economic development [9]. A similar categorization can be made from the perspective of local material flows.

Secondly, the environmental impact of different types of urban manufacturers could be empirically measured. This could establish whether environmental performance has any correlation with how 'locally embedded' urban manufacturers are.

### 5.1.3. Scale of Production and Environmental Impact

Our findings indicate that, even if urban manufacturers contribute to a circular economy, they tend to produce at a small scale, making their (positive) impact on the city as a whole relatively insignificant.

There are two potential ways to increase the impact of urban manufacturers-scaling up the production of existing individual facilities, or increasing the total amount of urban manufacturers in the city. The barriers for scaling up the production of individual urban manufacturers is relatively well explored and summarized in this paper. The issue of increasing the total amount of urban manufacturers in a city, which would entail creating infrastructure, institutions, and a support network for a thriving community of urban manufacturers, is relatively unexplored.

Further research could therefore involve answering the question, "what are the conditions (such as infrastructure, institutions, support network, policies) for creating a thriving circular urban manufacturing community?"

### 5.1.4. Industries Suitable for Circular Urban Manufacturing

Literature and interviewees on both circular cities and urban manufacturing have specified industries that are suitable for circular or urban manufacturing processes. Finding the overlaps between the specified industries from both topics therefore gives insight into which sectors can act as drivers to scaling up circular and urban manufacturing activity. The industries that were specified by both circular cities and urban manufacturing experts are: construction and demolition, fashion, bio-based products, and electronics [4,29,51,52,70,80].

Urban manufacturing experts give further insight into the *attributes* of products that are suitable for urban manufacturing. In order to compensate for high rental costs, urban manufacturers often make products of high value, such as products that are customized, niche [50,72,74], design-driven, and technology-driven (Interviewee E). Urban manufactured products also tend to be small (Interviewee C, E), have short life-times (Interviewee C, H), and are essential goods (Interviewee A, E). Table 5 below provides a summary of products mentioned by authors and experts, as well as their attributes that make them suitable to be manufactured in urban areas.

To summarize, the findings of this paper point towards four further research directions: the geographical scales of the circular economy, the local embeddedness of urban manufacturers from the perspective of material flows, strategies for scaling up the impact of circular urban manufacturers, and industries suitable for circular urban manufacturing. Learning from our findings, we do not advocate for urban manufacturing of *all* products or the localization of *all* resource flows—that would be unrealistic and unsustainable. Instead, we recommend further research to identify resources that are suitable to be recirculated at the city level, and products that are suitable for locally embedded supply chains.

**Table 5.** Summary of products and attributes suitable for circular urban manufacturing.

| | High-Value | Small | Design-Driven | Customized | Essential | Technology Driven | Niche | Short Life-Times | Perishable | Heavy |
|---|---|---|---|---|---|---|---|---|---|---|
| C&D [1] | x | | x | x | x | x | | | | x |
| Fashion | x | x | x | x | x | | | x | | |
| Healthcare | x | x | | x | x | x | x | | | |
| Food | x | x | | | x | | | x | x | |
| Electronics | x | x | x | | | x | x | | | |
| Diamonds | x | x | x | x | | | | | | |
| Furniture | x | | x | x | | | | | | |
| Plastic | | x | x | | | | | | | |

[1] Construction and demolition.

### 5.2. Limitations of Methodology

The research methods used in this paper come with a number of limitations, which are summarized in the paragraphs below.

### 5.2.1. Review of LCAs on Urban Manufacturing

Due to the limited availability of LCAs on urban manufacturing, this paper was only able to consider a limited number of LCAs (nine in total). The LCAs cover the production of food and consumer products, but other materials prioritized by circular economy literature, such as construction materials and electronics, were not included.

Moreover, the study focused on studying GHG emissions calculated by the LCAs, and not the other impact categories. This is because GHG emissions are the most commonly calculated impact category, making it possible to compare the different studies. However, this limits us from having a full understanding of the environmental impact of urban manufacturing, including its effects on human toxicity, air pollution, and water pollution.

Moreover, this paper did not take into account studies that used social LCA—a type of life cycle assessment that includes the social impacts along the supply chain of production processes. The societal costs and benefits of urban manufacturing have therefore not been considered during the review of LCAs. However, these costs and benefits were partially covered by interviews with experts, as well as the reviews of drivers and barriers of circular urban manufacturing.

### 5.2.2. Selection and Interviewing of Experts

During the selection process of interviewees for this paper, it was found that there are very few experts who have a deep understanding of both circular economy and urban manufacturing. This is unsurprising, since our paper was written precisely to address this research gap.

However, the lack of interviewees with expertise in both fields limits our certainty on the results of this paper. Most interviewees were able to provide in-depth insight to their respective field (CE or UM), but did not create a significant connection between the two topics. Therefore, we see our summary of drivers and barriers for circular urban manufacturing as a first attempt to connect these two fields in a coherent manner.

## 6. Conclusions

### 6.1. Summary of Research

The aim of this research was to establish whether urban manufacturing is a viable strategy for developing a circular economy in cities by answering two research questions: (1) Does urban manufacturing contribute to circular economy in cities, and how? and (2) What are the drivers and barriers to circular urban manufacturing?

The questions were answered through conducting a literature review of Life Cycle Assessments (LCAs) of urban manufacturing processes, interviews of experts within the fields of circular economy and urban manufacturing, and a literature review of drivers and barriers in both circular cities and urban manufacturing.

It was found that, while there is empirical evidence that urban manufacturing can reduce GHG emissions by reducing transportation distances and using waste as a resource, these claims come with a number of caveats. These caveats include the fact that transportation emissions contribute far less to GHG emissions than other production processes, that the scale of urban manufacturers is often too small to make an impact on the city as a whole, and that there are barriers to scaling up existing urban manufacturing activity.

This research then defined the drivers and barriers to circular urban manufacturing, and categorized them under the perspectives of "space", "people", and "flows". Spatial drivers and barriers are related to the availability of industrial land in cities, which depends on the willingness and ability of municipalities to protect industrial land. People-related drivers and barriers are related to urban manufacturing businesses' access to support networks, as well as their ability to scale up production while maintaining an urban location. Flow-related drivers and barriers are related to the availability and quality of local raw materials, including both municipal and industrial waste.

The conclusion of the research is that, while there is potential for urban manufacturing to contribute to circular cities, their current impact on the city as a whole is limited. Further research is needed to understand the conditions that allow urban manufacturers to scale up their production while remaining in the city.

### 6.2. Theoretical and Practical Contributions

This paper's theoretical contribution is that it addresses the research gaps between existing literature on urban manufacturing and circular economy in cities. In response to urban manufacturing literature, this paper questions the assumption that local production is more sustainable due to lower transportation emissions. In response to circular cities literature, this paper summarizes the space, people, and flow-related drivers and barriers of implementing circular urban manufacturing.

In terms of practical contributions, this paper provides insight for municipalities that are interested in transitioning to a circular economy by fostering a suitable environment for circular urban manufacturers.

### 6.3. Further Research

The results of this paper point to a number of directions for further research. More details can be found in the discussion section.

This paper has found that there is a need for a deeper understanding on the spatial scales of the circular economy. While there is an implicit assumption that limiting production systems to the city scale is more preferable to globalized production, this paper has found that this is not always the case. Given this finding, the next question to be investigated could be, "under what conditions is waste-to-resource conversion at a local scale more preferable to a national or global scale?"

This paper has found that research on urban manufacturing seems to implicitly assume that urban manufacturers are locally embedded, using local supply chains and producing for local consumer bases, while that this is not always true. Given this finding, it would be beneficial to categorize urban manufacturers according to how 'locally embedded' they

are—whether they utilize local, national, or global supply chains; and whether they sell to a local, national, or global consumer base.

This paper has summarized the drivers and barriers for individual urban manufacturers. However, the issue of creating a suitable environment for urban manufacturers, which would entail creating infrastructure, institutions, and a support network, is less explored. Further research could therefore involve answering the question, "what are the conditions (such as infrastructure, institutions, support network, policies) for creating a thriving circular urban manufacturing community?"

**Supplementary Materials:** The following are available online at https://www.mdpi.com/2071-105 0/13/1/23/s1, S1: Transcript of Interviews with Experts; S2: Interviews, Summary and Coding; S3: Literature, Summary and Coding; S4: LCAs of Urban Manufacturing Summary.

**Author Contributions:** Funding acquisition, D.P.; investigation, T.T.; methodology, T.T.; project administration, T.T.; supervision, D.P., B.G., and A.v.T.; writing—original draft, T.T.; writing—review and editing, T.T., D.P., B.G., and A.v.T. All authors have read and agreed to the published version of the manuscript.

**Funding:** This research received funding from the European Union's Horizon 2020 Research and Innovation Programme under grant agreement No 821479.

**Acknowledgments:** We would like to thank our interviewees for their time and effort—from being interviewed to reviewing the transcripts together. We are also very grateful for helpful discussions with researchers Birgit Hausleitner, Adrian Hill, as well as our project partners in the Pop-Machina project. Finally, we would like to thank Joan Huang for helping with the transcription of the interviews.

**Conflicts of Interest:** The authors declare no conflict of interest. The funders had no role in the design of the study; in the collection, analyses, or interpretation of data; in the writing of the manuscript, or in the decision to publish the results.

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
