# Peer review of "The Role of Urban Manufacturing for a Circular Economy in Cities"

_sustainability, doi:10.3390/su13010023_

Round 1

Reviewer 1 Report

The paper provides a valuable contribution to the understanding of the barriers and drivers to circular economy in cities with respect to urban manufacturing through literature review and expert interviews.

The boundaries of the research need clarification. On line 96, it is stated that the focus is on ‘techno cycle strategies such as reuse, refurbish, remanufacture, and recycle’. On line 117 and 124, agriculture and food are noted, and on line 215 the nine articles examined include food products. The supplementary material on LCA appears to include a majority of food products. On line 295, farmers and fertilizer are mentioned. To include food products under the definition of ‘circular urban manufacturing’ on lines 352-358, biological cycles would appear to be required to satisfy bullet point 3.

Line 95: The city scale is an example of macro level CE (Saidani et al. (2019) A taxonomy of circular economy indicators. Journal of Cleaner Production. 207, 542-559. doi 10.1016/j.jclepro.2018.10.014). The research presented here appears to examine the interaction of micro (manufacturer) and macro (city) scale CE – (re)-introducing circular urban manufacturing. This novelty could be more clearly stated.

The environmental focus is reduced to consideration of GHG emissions, comparing circular to traditional manufacturing (section 4.1). The limitations of LCA should be stated, possibly in the Discussion, including the relevance of S-LCA (social LCA).

Table 3 (Line 547) Unlike the drivers in Table 2, the barriers presented in Table 3 appear to be mostly barriers for urban start-ups in general. The barriers should be related more clearly to the specifics of circular urban manufacturing. In this respect, Kirchherr et al. (2018) (Barriers to the Circular Economy: Evidence from the European Union (EU) doi: 10.1016/j.ecolecon.2018.04.028) might provide guidance on how to frame these. For example, flows: limited availability of quality recycled material from local sources; lack of incentives to maintain critical materials in circularity locally. People: lack of circular product awareness and interest of consumers, etc. Space: lack of neighbourhood-friendly access to hubs for secondary material collection, sorting and distribution.

The limitations of the research should be discussed in terms of the methods used: adequacy of the literature on LCA, adequacy of coverage of topic in terms of interviewees.

Line 241 question 9 is not a clear question (language issue?). Please clarify.

Line 147 remove ‘s’ in ‘creates’

Line 184 ‘the’ in ‘the claims’

Line 302 advantage ‘of’ … economies of ‘scale’ missing

Line 374 missing word

Accept after minor revisions

Reviewer 2 Report

Comments to the Author

This paper has the potential to be very interesting, but at present, it requires some work to get a coherent comprehensive strategy to integrate both analyzed practical indicators in urban manufacturing and circular economy to track the improvement.  However, I think that some minor modifications and re-writing of the paper are needed to make the paper publishable.

I really find out conflictive that you referred a bibliometric analysis out of just 9 papers in the cross-section of both disciplines. You can maybe improve the scope including institutional reports, books and other scientific or business materials.

You need to explain better the use of push and full factors in the Circular Urban Manufacturing drivers setting. You understand the big picture when you get to table 2 but not before.

The controversy local/global should be better introduced and contextualized according to the objectives of your paper.

Confidentiality of the interviewees participants should be respected, but I think that for the aims of your study the including of information like country of origin and/or affiliation could be supply interesting insight to better contextualize the interventions of each participant.

Specific comments

In the lines 31-33 you need to give arguments that supports the assumptions you present therein.

In line 78, when you mention closing loops as the main aim of circular economy, the current existing literature also include extending loops and intensifying loops as the most important together with closing loops.

Acknowledgment section should be rewritten, you should adapt that to your paper.

Some convoluted paragraphs need to be rewritten: lines 154-159, 184, 190-191 and 323

Reviewer 3 Report

This is an interesting article, which deals with a contemporary issue of the interlinkages between urban manufacturing and circular economy. However, there are some areas where the paper needs to be improved upon.

1) Introduction:

The authors should explain in detail the concept of circular economy and why it is relevant to urban manufacturing.

2) Review of literature

The linkages between circular economy and sustainability and the three categorisations vis tradeoffs, beneficial and conditional need to be explained more fully.

3) Methodology

The selection of participants and categories for coding need to be explained further.

4) Conclusion

The conclusion section should elucidate on the theoretical and practical contributions of the study, the limitations of the study as well as avenues for further research.

Round 2

Reviewer 3 Report

I have gone through the revisions. I am satisfied with the revisions undertaken.